# Combined Hyperthermic Intraperitoneal Chemotherapy and Normothermic Intraperitoneal Chemotherapy Long-Term After Interval Cytoreduction in Ovarian Cancer: A Phase I Clinical Trial (BICOV1)

**DOI:** 10.3390/cancers17121957

**Published:** 2025-06-12

**Authors:** Alida González-Gil, Elena Gil-Gómez, Vicente Olivares-Ripoll, Álvaro Cerezuela Fernández de Palencia, Jerónimo Martínez-García, Domingo Sánchez-Martínez, Alberto Rafael Guijarro-Campillo, Pedro Antonio Cascales-Campos

**Affiliations:** 1Peritoneal Carcinomatosis and Sarcomas Unit, Department of Surgery, Hospital Universitario Virgen de la Arrixaca, IMIB-Arrixaca, 30120 Murcia, Spain; alidagonzalezgil@gmail.com (A.G.-G.); elenagilgomez@hotmail.com (E.G.-G.); vicenteolivaresripoll@gmail.com (V.O.-R.); alvaro.cerezuela@gmail.com (Á.C.F.d.P.); 2Department of Surgery, University of Murcia, 30100 Murcia, Spain; 3Department of Medical Oncology, Hospital Universitario Virgen de la Arrixaca, IMIB-Arrixaca, 30120 Murcia, Spain; jeronimo@seom.org (J.M.-G.); domingoa.sanchez@carm.es (D.S.-M.); 4Department of Gynecologic Oncology, Hospital Universitario Virgen de la Arrixaca, IMIB-Arrixaca, 30120 Murcia, Spain; argc777@gmail.com

**Keywords:** ovarian cancer, intraperitoneal chemotherapy, HIPEC, NIPEC-LT, BICOV-1 trial

## Abstract

Ovarian cancer, the deadliest gynecologic malignancy, often presents with peritoneal dissemination. Complete cytoreduction and platinum-based chemotherapy sensitivity are key prognostic factors, yet over 50% of patients relapse due to residual microscopic disease. Intraperitoneal chemotherapy, including normothermic long-term (NIPEC-LT) and hyperthermic (HIPEC) approaches, targets this issue. NIPEC-LT improves survival but is limited by toxicity and catheter issues, while HIPEC, effective in interval cytoreductive surgery, is a single treatment. The BICOV-1 trial, a phase I study, evaluates combining HIPEC and postoperative NIPEC-LT for feasibility, safety, and outcomes. Primary endpoints are treatment completion and morbidity; secondary endpoints include disease-free survival, overall survival, and quality of life. This trial will guide future studies on intensified intraperitoneal therapy for ovarian cancer.

## 1. Introduction

Ovarian cancer remains the leading cause of death from gynecologic malignancies. At the time of initial diagnosis, more than two-thirds of patients present with peritoneal dissemination [1,2]. Complete cytoreduction during primary surgery, combined with tumor sensitivity to platinum-based systemic chemotherapy, represent the two most significant prognostic factors. However, even when macroscopic complete cytoreduction is achieved, over half of the patients experience relapse during follow-up, predominantly with intraperitoneal localization, due to residual microscopic disease [3]. The use of intraperitoneal chemotherapy effectively targets this microscopic component, with normothermic intraperitoneal chemotherapy long-term (NIPEC-LT) or hyperthermic intraperitoneal chemotherapy (HIPEC) being the most employed modalities in ovarian cancer [4,5].

The efficacy of normothermic intraperitoneal chemotherapy administered as part of long-term treatment (NIPEC-LT) in advanced ovarian cancer has been investigated in three prospective, randomized clinical trials conducted by the Gynecologic Oncology Group (GOG) [6,7,8]. Among these, the GOG 172 trial [8] provided pivotal data supporting the clinical benefit of intraperitoneal (IP) chemotherapy in patients with stage III epithelial ovarian cancer. In this trial, patients who had undergone optimal cytoreductive surgery were randomized to receive either standard intravenous (IV) chemotherapy or a combined regimen of IV paclitaxel followed by IP cisplatin and paclitaxel. The results demonstrated a significant improvement in overall survival (OS) in the IP arm, with a median OS of 65.6 months compared to 49.7 months in the IV-only group. Progression-free survival (PFS) also improved in the IP group, with a median of 23.8 months versus 18.3 months in the IV group. These outcomes support the rationale that direct peritoneal drug delivery allows for higher local concentrations of chemotherapy, potentially enhancing the eradication of microscopic residual disease and improving long-term survival [8]. Further analysis of a subgroup of patients who achieved complete cytoreduction—defined as no gross residual disease—revealed even more substantial benefits. In this population, disease-free survival (DFS) reached a median of 60.4 months (range: 36.9–not applicable [NA]), and OS extended to 127.6 months (range: 84.7–NA) [9], suggesting that the effectiveness of IP chemotherapy may be closely linked to surgical outcomes and residual tumor burden.

Despite these promising results, NIPEC-LT has not been established as a standard of care. Limitations include increased systemic toxicity, catheter-related complications, and reduced tolerability, which collectively contributed to a lower completion rate of planned therapy in the IP arm of GOG 172. Patients receiving IP chemotherapy reported significantly greater toxicity, particularly neurotoxicity, abdominal pain, and gastrointestinal symptoms, leading to lower quality of life scores during treatment and recovery [8].

In an effort to further evaluate the role of IP chemotherapy in the era of targeted therapies, the GOG 252 trial was conducted [10]. This study introduced bevacizumab, an anti-angiogenic agent, across all arms and compared three treatment regimens: IV chemotherapy plus bevacizumab, IP cisplatin plus bevacizumab, and IP carboplatin plus bevacizumab. Contrary to expectations, the incorporation of bevacizumab did not lead to improvements in DFS or OS. Median DFS was 24.9 months in the IV group, 26.2 months in the IP cisplatin group, and 27.4 months in the IP carboplatin group. Similarly, no statistically significant differences in OS were observed among the treatment arms. These findings diminished enthusiasm for the routine use of NIPEC-LT in newly diagnosed advanced ovarian cancer, particularly in light of its toxicity profile and lack of incremental benefit when combined with bevacizumab [10].

Intraoperative intraperitoneal chemotherapy associated with hyperthermia (HIPEC) has become strongly positioned in the management of ovarian cancer after the publication of the OVHIPEC-1 clinical trial [11], which evaluated its role in patients undergoing interval cytoreduction following neoadjuvant systemic chemotherapy. This pivotal phase III randomized trial by van Driel et al., including 245 patients with stage III epithelial ovarian cancer, demonstrated that adding HIPEC with cisplatin (100 mg/m^2^ for 90 min at 40 °C) significantly improved both progression-free survival (PFS) and overall survival (OS). Median PFS increased from 10.7 to 14.2 months, and median OS from 33.9 to 45.7 months in the HIPEC group, without increased morbidity or deterioration in quality of life.

Supporting this, Cascales-Campos et al. conducted a prospective, randomized phase III trial involving 71 patients with peritoneal carcinomatosis from ovarian cancer treated with neoadjuvant chemotherapy [12]. Patients were randomized to undergo cytoreductive surgery (CRS) alone or CRS plus HIPEC (cisplatin 75 mg/m^2^, 60 min at 42 °C). After a median follow-up of 32 months, median DFS was 18 months in the HIPEC group versus 12 months in the control group (HR 0.12; 95% CI 0.02–0.89; *p* = 0.038). Median OS was also longer: 52 months in the HIPEC group versus 45 months in the control group. No significant differences in morbidity (45.7% vs. 58.3%) or mortality (2.9% vs. 2.8%) were observed, and quality of life remained unaffected by the addition of HIPEC.

Lim et al. [13] also conducted a single-blind randomized clinical trial in South Korea, including 184 patients with stage III or IV ovarian cancer with residual disease <1 cm after surgery. Patients received either HIPEC (cisplatin 75 mg/m^2^, 90 min at 41.5 °C) or no HIPEC. While the overall population did not show significant differences in PFS or OS, the subgroup of patients undergoing interval cytoreductive surgery after neoadjuvant chemotherapy showed clear benefit: median PFS was 17.4 months in the HIPEC group vs. 15.4 months in the control group (HR 0.60; 95% CI 0.37–0.99; *p* = 0.04), and median OS was 61.8 months vs. 48.2 months, respectively (HR 0.53; 95% CI 0.29–0.96; *p* = 0.04). No significant increase in adverse events was reported.

A meta-analysis by Filis et al. [5] further reinforced these findings, with pooled data showing hazard ratios of 0.56 for OS and 0.61 for PFS, favoring the use of HIPEC, indicating substantial risk reduction in recurrence and mortality. Furthermore, interval cytoreductive surgery with HIPEC has been shown to be cost-effective in stage III–IV ovarian cancer [14,15]. In summary, these results confirm the potential of HIPEC as an effective and safe adjunct to cytoreductive surgery in both primary and recurrent ovarian cancer settings with comparable morbidity, mortality, and quality of life. Currently, HIPEC is included in clinical guidelines for patients undergoing iCRS [16,17]. Furthermore, the CHIPOR trial—focused on recurrent ovarian cancer—showed that HIPEC extended median PFS from 10.2 to 13.1 months without increasing complications, suggesting a role for HIPEC beyond initial treatment [18].

One unresolved limitation of HIPEC, frequently criticized, is that its benefit relies on a single administration. Implementing a cyclic intraperitoneal treatment regimen would require the placement of an abdominal catheter for multiple NIPEC-LT cycles post-discharge. To date, no prior studies in the literature have combined NIPEC-LT following HIPEC. However, the combination of these two intraperitoneal treatment modalities appears rational, as their complications do not overlap temporally, and both have independently demonstrated survival benefits. The BICOV-1 clinical trial, a prospective, non-randomized phase I study, will evaluate the combination of HIPEC and NIPEC-LT following iCRS.

## 2. Material and Methods

The inclusion and exclusion criteria are detailed in Table 1. The trial flowchart is shown in Figure 1.

### 2.1. Primary Outcome

The primary objective is to assess the safety of combining NIPEC-LT and HIPEC after cytoreduction by analyzing associated morbidity and mortality, as well as the percentage of patients completing the full intraperitoneal treatment protocol. Adverse events will be classified according to the National Cancer Institute’s Common Terminology Criteria for Adverse Events (NCI-CTCAE, version 5.0) [19]. All complications occurring within the first 90 days post-surgery, including procedure-related mortality and those following NIPEC-LT treatment, will be recorded.

### 2.2. Secondary Outcomes

The secondary objectives include the analysis of DFS and OS (in months) at 1, 3, and 5 years, alongside treatment-associated quality-of-life parameters. Quality of life will be assessed using the EORTC QLQ-C30 questionnaire [20] with the ovarian cancer-specific module (OV-28), the EuroQoL Quality of Life Scale (EQ-5D) [21], and the State-Trait Anxiety Inventory (STAI) [22]. These assessments will be conducted pre-surgery (baseline) and repeated at 3, 6, and 12 months post-surgery.

### 2.3. Sample Size and Data Analysis

For a 95% confidence level, a 15% prevalence of severe complications, and a 5% margin of error, an initial sample size of 22 patients was calculated. Accounting for a 30% loss rate—based on over 15 years of experience from participating groups and the potential for failure to achieve CC-0 cytoreduction or the need for digestive anastomosis—an initial recruitment of 32 patients was deemed necessary.

The study will use frequency tables to detect errors and to ensure accurate variable entry, verifying consistency with the total number of enrolled patients. For quantitative variables, the standard deviation will be calculated, and percentile distributions will be analyzed to select appropriate statistical tests. Contingency tables will identify duplicates or inconsistencies. After defining the intention-to-treat (ITT) and per-protocol (PP) populations, the database will be locked in statistical analysis. Primary variables will be described using measures such as the mean, standard deviation, and median for numerical variables and frequency distributions for categorical variables. Survival analysis will be conducted using Kaplan–Meier curves, with the log-rank or Wilcoxon method, to compare survival distributions.

### 2.4. Treatment Strategy

#### 2.4.1. Surgical Procedure

Following a wide midline laparotomy, the extent of peritoneal disease is assessed using the Peritoneal Cancer Index (PCI), a validated quantitative scoring system that ranges from 0 to 39 and serves as a key prognostic and therapeutic guide in patients with peritoneal carcinomatosis [23]. The PCI enables a systematic mapping of tumor burden by dividing the abdominal cavity into 13 regions and assigning lesion size scores in each region. All patients undergo a standardized operative procedure designed to maximize the efficacy of cytoreductive surgery (CRS) and ensure uniformity in surgical approach. Surgical procedures routinely performed as part of this protocol include total hysterectomy, bilateral salpingo-oophorectomy, and complete omentectomy, as these sites are frequently involved in advanced-stage ovarian cancer. The outcomes of the cytoreductive surgery are assessed using the Completeness of Cytoreduction Score (CCS), a standardized classification system that reflects the amount of residual tumor left behind after surgery. A CCS of 0 (CC-0) indicates complete macroscopic resection, meaning no visible disease remains. In those patients who meet the inclusion criteria and are selected to continue with the study protocol, no digestive anastomosis is performed during the primary surgical intervention. Importantly, achieving a CC-0 cytoreduction is a prerequisite for proceeding to the HIPEC phase in this protocol. The rationale for this requirement is based on evidence suggesting that the efficacy of HIPEC is significantly enhanced in the absence of macroscopic disease, where the chemotherapeutic agent can directly interact with microscopic residual tumor cells [24].

#### 2.4.2. HIPEC Protocol and Intraperitoneal Catheter Management

After complete cytoreduction, HIPEC will be administered with cisplatin (100 mg/m^2^, 42 °C, 90 min). Intravenous sodium thiosulfate (9 g/m^2^ diluted in 200 mL as a bolus, followed by a continuous infusion of 12 g/m^2^ in 1000 mL over 6 h) is protocolized to prevent nephrotoxicity. This study adopts a closed HIPEC technique with internal agitation via CO₂ recirculation. Upon HIPEC completion, a 15-Fr intraperitoneal “port-a-cath” catheter will be placed for subsequent NIPEC-LT administration. The catheter head for puncture and infusion will be positioned on the anterior thoracic wall at the level of the lower right ribs, tunneled along the right axillary line, and introduced into the peritoneal cavity at the umbilical level, extending to the right iliac fossa (Appendix A). Prior to adjuvant treatment initiation, an imaging test (plain abdominal X-ray) will confirm the correct catheter positioning. Upon completion of intraperitoneal treatment, the catheter will be removed.

#### 2.4.3. NIPEC-LT Protocol

NIPEC-LT treatment will be determined by the multidisciplinary team, ideally between 30 and 60 days post-surgery. Patients will receive 4 cycles of NIPEC-LT every 21 days, following a regimen adapted from Armstrong and modified by the Spanish Ovarian Cancer Research Group (GEICO): intravenous paclitaxel 175 mg/m^2^ (day 1) with intraperitoneal cisplatin 100 mg/m^2^ (day 2) and intraperitoneal paclitaxel 60 mg/m^2^ (day 8) [25]. Toxicity management will be adjusted based on the type and grade of morbidity. For grade 1 toxicity, treatment will continue with enhanced symptomatic support. For grade 2–3 toxicity, the next administration will be delayed, with consideration of dose reduction (20% reduction in paclitaxel dose and 1 area under the curve [AUC] reduction in carboplatin dose). For grade 4 toxicity, intraperitoneal treatment will be permanently discontinued, except in cases of hematologic toxicity where support with filgrastim or erythropoietin is indicated and effective.

### 2.5. Ethics

The BICOV-1 trial protocol has been approved by the Clinical Research and Ethics Committee of Hospital Clínico Universitario Virgen de la Arrixaca (IMIB-BICOV-2022-01) and authorized by the Spanish Agency of Medicines and Medical Devices (AEMPS). It is registered in the Spanish Clinical Studies Registry (ReEC, 2022-502691-23-00), as well as in the EudraCT (2022-001107-41) and ClinicalTrials (NCT06902467) databases.

The study will be conducted in Spanish hospitals in accordance with current regulations (Royal Decree 957/2020) and the ethical principles of the Declaration of Helsinki [26]. Patients will provide written informed consent after receiving detailed verbal and written explanations. Participation is voluntary and can be withdrawn at any time. Data confidentiality will be ensured by coding patient identities, restricting access to authorized personnel, and complying with the General Data Protection Regulation (GDPR 2016/679) and Spanish data protection legislation (Organic Law 3/2018).

Study data will be securely stored at participating hospitals, with access limited to investigators, regulatory authorities, and auditors. The designated Institutional Review Board will oversee regulatory approvals, study monitoring, and database updates. All records will be archived per ICH guidelines for good clinical practice and European Directive 2005/28/EC. Investigators will safeguard confidential information, restricting access to authorized personnel only.

## 3. Discussion

The goal of intraperitoneal chemotherapy in ovarian cancer is to target the microscopic disease component that remains undetectable to the surgeon after macroscopic cytoreduction. In patients with newly diagnosed peritoneal dissemination from ovarian cancer, NIPEC-LT has shown efficacy but has not been established as a standard treatment [6,7,8,9,10]. Similarly, NIPEC-LT appears to improve outcomes in patients undergoing iCRS. Meanwhile, HIPEC has demonstrated improved DFS and OS in various clinical trials favoring intraperitoneal treatment [11,12,13]. However, no protocols in the literature combine these two intraperitoneal chemotherapy modalities, which, despite offering clear prognostic advantages, could be applied complementarily in the same patient without temporally overlapping complications.

HIPEC, administered during surgery, derives its benefit from a combination of mechanical (continuous perfusion flow removes some microscopic components), physical (hyperthermia), and pharmacological (cytostatic drugs) effects on residual microscopic disease. However, HIPEC is limited to a single administration. Combining it with NIPEC-LT would not only enable a cyclic intraperitoneal regimen but also leverage HIPEC-induced biological changes that may enhance the efficacy of NIPEC-LT. HIPEC induces multiple alterations in the tumor microenvironment: At the cellular level, hyperthermia and chemotherapy increase apoptosis due to changes in cell membrane integrity and intracellular protein damage. This is accompanied by elevated expression of heat shock proteins (HSPs), such as Hsp70 and Hsp90, which promote apoptosis and serve as immunogenic signals that enhance the antitumor immune response [27].

From a molecular perspective, HIPEC modulates signaling pathways that influence the treatment response. It has been reported to inhibit homologous recombination DNA repair, sensitizing tumor cells to drugs such as cisplatin [28]. Additionally, significant activation of the tumor necrosis factor-alpha (TNF-α)/nuclear factor kappa-light-chain-enhancer of activated B cells (NF-κB) pathway has been observed in patients with favorable HIPEC responses, suggesting a key role in tumor sensitization to subsequent chemotherapy.

Another critical effect of HIPEC is the restructuring of the tumor microenvironment. Increased infiltration of immune cells, particularly activated CD8+ T lymphocytes, indicates enhanced antitumor immunity post-treatment [27]. Similarly, programmed cell death protein 1 (PD-1) expression on T cells increases after HIPEC, potentially reflecting both immune activation and tumor-mediated inhibition mechanisms [28].

The cellular and molecular changes induced by HIPEC may thus enhance the efficacy of postoperative intraperitoneal chemotherapy through several mechanisms. First, HIPEC-induced apoptosis reduces the residual tumor burden, allowing subsequent chemotherapy to target a smaller tumor population. Second, inhibition of DNA repair in HIPEC-pretreated cells augments sensitivity to postoperative chemotherapeutic agents, particularly platinum-based drugs [27]. Furthermore, hyperthermia increases cellular permeability, improving drug uptake and optimizing cytotoxic effects [29]. Combined with HIPEC-induced tumor microenvironment remodeling, this facilitates better diffusion of chemotherapeutic agents into affected peritoneal tissue [28].

Finally, activation of an antitumor immune response mediated by HSPs and cytotoxic T-cell infiltration creates a hostile environment for tumor progression. Immunomodulation could play a pivotal role in reducing recurrence and improving prognosis in patients treated with HIPEC and postoperative intraperitoneal chemotherapy [30].

## 4. Conclusions

HIPEC induces cellular and molecular changes that may enhance the effectiveness of postoperative intraperitoneal chemotherapy. Through apoptosis induction, DNA repair inhibition, increased cellular permeability, and antitumor immune activation, HIPEC optimizes the impact of subsequent chemotherapy. These findings underscore HIPEC’s relevance as a therapeutic strategy in peritoneal carcinomatosis and justify its integration into clinical protocols to improve long-term oncologic outcomes. Given that the complications of both modalities do not overlap temporally, and both have demonstrated survival benefits, the proposed intensification of intraperitoneal treatment in this protocol is warranted. Data from this trial will serve as a foundation for designing future phase II and III clinical trials comparing the administration or omission of NIPEC-LT following iCRS with HIPEC.

## Figures and Tables

**Figure 1 cancers-17-01957-f001:**
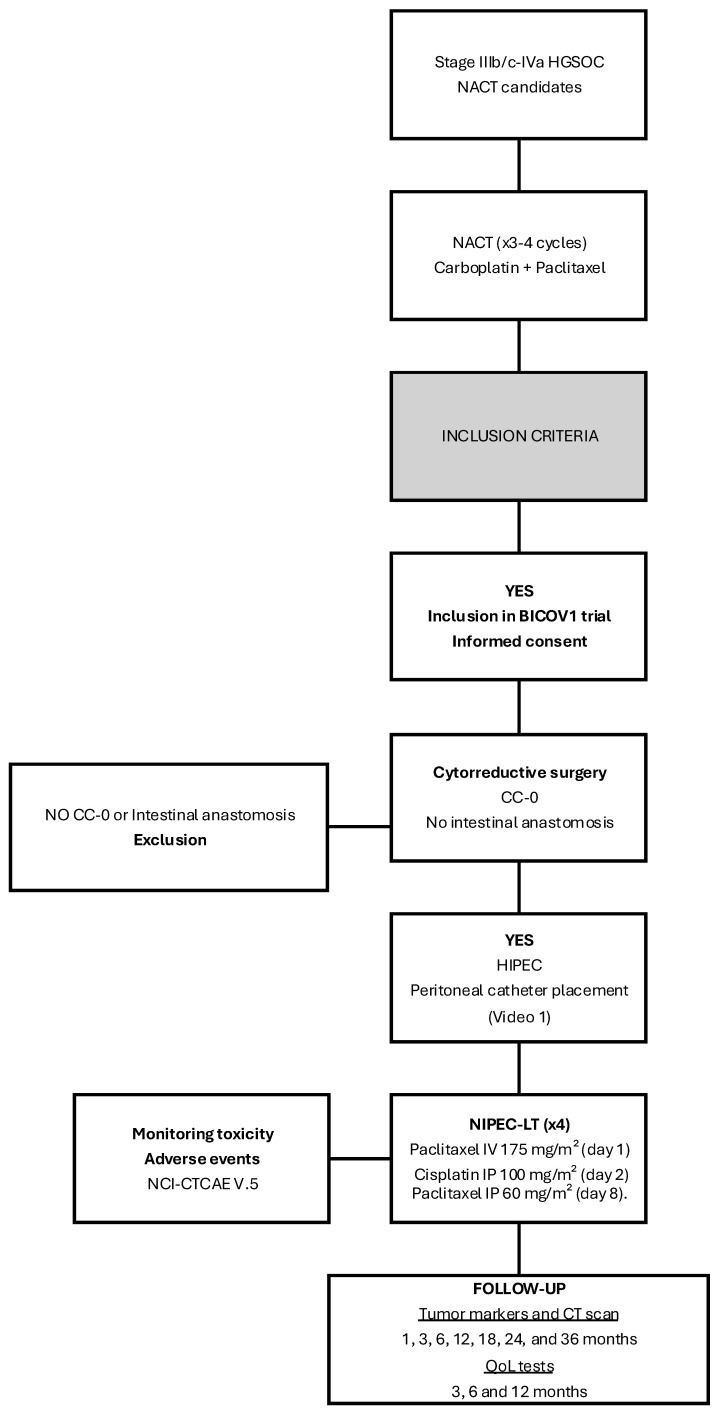
Flow chart.

**Table 1 cancers-17-01957-t001:** BICOV1 clinical trial. Inclusion and exclusion criteria.

INCLUSION CRITERIA
Women aged 18–70 years with FIGO stage IIIB–C or IV high-grade epithelial ovarian cancer. Disease confined to the peritoneum without distant metastases at evaluation. Karnofsky Performance Score >70 or ECOG Performance Status ≤2. Adequate liver, kidney, and bone marrow function per protocol-defined parameters. No serious comorbidities contraindicating major surgery. Confirmed negative pregnancy test in women of childbearing potential. Completion of 3–4 cycles of neoadjuvant systemic chemotherapy. Complete cytoreduction with no visible residual disease. No requirement for digestive anastomosis during cytoreductive surgery. Provision of signed informed consent.
EXCLUSION CRITERIA
Tumor progression following neoadjuvant systemic chemotherapy. Extraperitoneal disease identified on re-evaluation after neoadjuvant therapy. Inability to achieve complete cytoreduction Requirement intestinal resection and anastomosis during cytoreductive surgery Documented active infection. Enrollment in another clinical trial within the last 30 days. Current pregnancy or breastfeeding. Allergy to platinum or paclitaxel components. Failure to provide signed informed consent.

## Data Availability

To access the data, please contact the corresponding author, cascalescirugia@gmail.com.

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
