# Peer review of "Combined Hyperthermic Intraperitoneal Chemotherapy and Normothermic Intraperitoneal Chemotherapy Long-Term After Interval Cytoreduction in Ovarian Cancer: A Phase I Clinical Trial (BICOV1)"

_cancers, 2025, doi:10.3390/cancers17121957_

Round 1

Reviewer 1 Report

Comments and Suggestions for Authors

The manuscript focus  on role of  combined mechanical, physical and pharmacological therapy in ovarian cancer, one of the most common cancer-associated cause of death in women. The Authors presented the clear introduction, clear inclusion and exclusion criteria, the primary and secondary outcomes are described in a clear and appropriately reasoned manner, as well as treatment strategy is detailed decribed. 

The section "Sample size and data analysis" Authors are presenting the the methodology, which includes both intention-to-treat (ITT) and per-protocol (PP) analyses, which enhances the reliability of the results. 

Overall, the data analysis protocol reflects a high standard of statistical planning and adherence to best practices in clinical research.

The Discussion makes a reasonable effort to draw attention to the potential  clinical relevance of combined HIPEC and NIPEC-LT novel therapy in ovarian cancer treatment.

Major issues:

It is recommended that the manuscript be supplemented with clinical data obtained from the initial patient cohort to enhance its scientific value and provide preliminary insights into the study outcomes.

Author Response

Dear reviewer

First of all, thank you very much for your comments. We agree that including preliminary clinical data could improve the article, but you should keep in mind that this is a protocol article and therefore we do not yet have enough data to be able to offer at least a significant series.

Reviewer 2 Report

Comments and Suggestions for Authors

This paper presents a protocol of BICOV1 study which evaluates a combination of HIPEC and NIPEC-LT in advanced ovarian cancer. I have the following comments: 

  1. Will this be multicentric international trial? How many hospitals will be included?
  2. How many patients do authors plan to include in this study? How many patients are diagnosed with advanced ovarian cancer in the authors' institution each year?
  3. How will the paients be monitored for toxicity?

Author Response

Dear reviewer, we would first like to thank you for your comments. This is a single-center clinical trial to assess the feasibility of using the combined intraperitoneal chemotherapy protocol (HIPEC with NIPEC-LT). This clinical trial will be used to plan a prospective, randomized clinical trial, which will be multicenter if the results of this trial are favorable.
For a 95% confidence interval, a prevalence of serious complications of 15%, and a margin of error of 5%, an initial sample size of 22 patients was calculated. Considering a 30% loss rate, based on more than 15 years of experience in the participating groups and the possibility of not achieving CC-0 cytoreduction or performing a digestive anastomosis, an initial recruitment of 32 patients was estimated to be necessary.

Adverse events will be classified according to the National Cancer Institute toxicity criteria (NCI-CTCAE, version 5.0). All complications occurring during the first 90 days from the date of surgery, including procedure-related mortality, and subsequently after treatment with NIPEC-LT will be recorded. Management of NIPEC-LT-associated toxicity will be adjusted according to the type and degree of morbidity. In patients with grade 1 toxicity, treatment will be continued, adjusting adjunctive symptomatic support. In cases of grade 2-3 toxicity, the next administration will be delayed, considering a dose reduction (20% of the paclitaxel dose and 1 AUC of the carboplatin dose). In patients with grade 4 toxicity, intraperitoneal treatment will be permanently discontinued, except in cases of hematologic toxicity in which support with filgrastim or erythropoietin is indicated and effective.

Currently, our center is a regional reference center for the treatment of complex oncological pathologies and, specifically, to answer your question, we evaluate and treat approximately 100 patients with ovarian cancer at different stages per year.

Reviewer 3 Report

Comments and Suggestions for Authors

This is an interesting study. However, considering that intraperitoneal chemotherapy seems most useful in patients debulked to ≤1 cm with stage III (in general) epithelial ovarian cancer, and even in earlier-stages, you should further justify the inclusion criteria of your study.  

Author Response

Dear reviewer,

Thank you very much for your comment. First, we would like to address the criterion of less than 1 cm. In the treatment of malignant diseases of the peritoneal surface, we use the Sugarbaker Peritoneal Cancer Index (PCI), in which we apply HIPEC with curative intent only to those patients with a complete macroscopic resection, or with a macroscopic tumor residue smaller than 2.5 mm. This maximum threshold of 2.5 mm has been shown to be effective in intraoperative intraperitoneal chemotherapy, as larger sizes do not ensure the efficacy of the therapy.

We have deliberately restricted the inclusion criteria regarding surgical outcomes to those patients with complete macroscopic resections, as only in these cases can the true role of a therapy aimed at microscopic disease be assessed. You suggest a threshold of less than 1 cm, but in fact, between 2.5 mm and 10 mm, our group does not consider the use of HIPEC appropriate.

Round 2

Reviewer 1 Report

Comments and Suggestions for Authors

Thank you, I accept the article in current form.